environmental science

carbon source, compound-specific stable isotope analysis of amino acids, lipid biomarker, marine mammal, phytoplankton, sea ice algae

**Author for correspondence:**
David J. Yurkowski
e-mail: dyurkowski1@gmail.com,
David.yurkowski@dfo-mpo.gc.ca

# Atlantic walrus signal latitudinal differences in the long-term decline of sea ice-derived carbon to benthic fauna in the Canadian Arctic

David J. Yurkowski[1], Thomas A. Brown[2], Paul J. Blanchfield[1] and Steven H. Ferguson[1]

[1]Fisheries and Oceans Canada, Winnipeg, Manitoba, R3T 2N6, Canada
[2]Scottish Association for Marine Science, Oban PA37 1QA, UK

DJY, 0000-0003-2264-167X; TAB, 0000-0003-0322-474X; PJB, 0000-0003-0886-5642;
SHF, 0000-0002-3794-0122

Climate change is altering the biogeochemical and physical characteristics of the Arctic marine environment, which impacts sea ice algal and phytoplankton bloom dynamics and the vertical transport of these carbon sources to benthic communities. Little is known about whether the contribution of sea ice-derived carbon to benthic fauna and nitrogen cycling has changed over multiple decades in concert with receding sea ice. We combined compound-specific stable isotope analysis of amino acids with highly branched isoprenoid diatom lipid biomarkers using archived (1982–2016) tissue of benthivorous Atlantic walrus to examine temporal trends of sea ice-derived carbon, nitrogen isotope baseline and trophic position of Atlantic walrus at high- and mid-latitudes in the Canadian Arctic. Associated with an 18% sea ice decline in the mid-Arctic, sea ice-derived carbon contribution to Atlantic walrus decreased by 75% suggesting a strong decoupling of sea ice-benthic habitats. By contrast, a nearly exclusive amount of sea ice-derived carbon was maintained in high-Arctic Atlantic walrus (98% in 1996 and 89% in 2006) despite a similar percentage in sea ice reduction. Nitrogen isotope baseline or the trophic position of Atlantic walrus did not change over time at either location. These findings indicate latitudinal differences in the restructuring of carbon energy sources used by Atlantic walrus and their benthic prey, and in turn a change in Arctic marine ecosystem functioning between sea ice–pelagic–benthic habitats.

## 1. Introduction

Arctic marine environments are fuelled by recurrent influxes of two distinct sources of carbon that are driven by the seasonal progression of light and open water availability which varies by latitude [1]. First, the sea ice algal bloom marks the transition from winter to spring and provides a pulse of 3–60% of total annual primary production from low-latitude to high-latitude Arctic marine systems [2,3]. Second, the sea ice algal bloom is subsequently followed by a pronounced summer phytoplankton bloom that provides the majority of primary productivity to the Arctic marine environment [3]. Climate warming is increasing the intensity of phytoplankton blooms in relation to thinning sea ice, a longer growing season, an influx of new nutrients, and warmer ocean temperatures which has also facilitated a poleward shift of more temperate-associated species leading to a reorganization of the food web [4–9]. Moreover, the structure of rocky-bottom Arctic communities are undergoing ecological regime shifts with prominent increases in both macroalgal cover and abundance of benthic invertebrates [10]. The seasonal influxes of both primary productivity sources to Arctic marine waters often exceed the retentive

capacity of pelagic consumers leading to high amounts of allochthonous organic carbon being transported to seafloor habitats [11,12]. This sympagic (ice algae)-pelagic-benthic coupling drives the energy flow between surface and benthic energy compartments within Arctic shelf habitats and forms the foundation of Arctic ecosystem functioning [13,14]. However, little is known about how the contribution of sea ice-derived and phytoplankton-derived carbon to benthic fauna (i.e. sympagic–pelagic–benthic coupling) has changed over multiple decades with receding sea ice.

Here, we use Atlantic walrus (*Odobenus rosmarus rosmarus*) to test for changes in the relative importance of sea ice algae versus phytoplankton carbon sources that fuel benthic communities. Atlantic walruses are central place foragers that are stenophagous and reside near shallow (less than 80 m) bivalve-rich benthic communities, their main prey [15,16], where they consume approximately 60 kg of filter-feeding bivalves per day, or approximately 3–6% of their body weight [17]. Two populations of Atlantic walrus occur in Canadian waters, a central/low-Arctic population and a high-Arctic population [18]. These populations are presently subdivided into seven management units [19,20]. In two of these management units, northern Foxe Basin and Baffin Bay near Grise Fiord, Nunavut, Canada, Atlantic walruses over-winter in polynyas near their summer feeding grounds within northern Foxe Basin (approx. 69° N) and Jones Sound (approx. 76° N), respectively [20]. The specialized feeding ecology and year-round restricted movement of Atlantic walrus in each of these geographically separated areas (Jones Sound is approx. 1000 km north of Foxe Basin) provide a unique opportunity to investigate spatio-temporal variation in carbon energy dynamics of Arctic benthic environments.

We used a novel biochemical approach by coupling source-specific highly branched isoprenoid diatom lipid bio-markers with compound-specific stable carbon and nitrogen isotope values of individual amino acids of archived walrus tissues from Jones Sound over an 11-year time period (1996–2006) and northern Foxe Basin over a 35-year time period (1982–2016). We investigated how annual carbon contributions from sea ice algae and phytoplankton differ between the high- and mid-Arctic benthic habitats in association with temporal and latitudinal variation in declining sea ice cover. Highly branched isoprenoids assess carbon source partitioning between sea ice algae and phytoplankton in Arctic environments, and have been used extensively in marine mammals [21–24]. Compound-specific stable isotope analysis of individual amino acids is a powerful tool to discern relative influences of the carbon and nitrogen isotope composition at the base of the food web and trophic fractionation on consumer $\delta^{13}C$ and $\delta^{15}N$ values to then infer consumer carbon source use and diet [25–27].

The $\delta^{13}C$ of essential amino acids (e.g. phenylalanine, a source amino acid) do not change between diet and consumer and therefore represent the $\delta^{13}C$ baseline of the carbon source [26], such as between the more $^{13}C$-enriched sea ice-derived carbon and the more $^{13}C$-depleted phytoplankton-derived carbon. For $\delta^{15}N$, the differentiation between source and trophic amino acids (e.g. glutamic acid) can be used to estimate the trophic position of a consumer from a single sample while taking into account the $\delta^{15}N$ of the baseline [28]. Source amino acids show very little isotopic fractionation between baseline and top predator $\delta^{15}N$ values whereas trophic amino acids show significant isotopic fractionation

with each trophic step [25,28]. Therefore, source amino acids provide a proxy for nitrogen cycling at the primary producer level who acquire essential nitrates from seawater and can be used to detect temporal changes in Arctic circulation dynamics [29].

The objectives of this study were to (1) determine whether the contribution of sea ice-derived carbon to Atlantic walrus has changed over time, (2) examine temporal changes in $\delta^{15}N$ of the baseline and the trophic position of Atlantic walrus and (3) investigate latitudinal variation in objectives (1) and (2) associated with summer sea ice concentration and sea ice breakup date between Jones Sound and northern Foxe Basin. We hypothesized that Atlantic walrus from the more-northern ice-covered waters of Jones Sound will have a higher contribution of sea ice-derived carbon in their tissues than individuals who reside in the more-southern Foxe Basin area as a result of higher summer sea ice concentrations. In addition, Atlantic walrus from Jones Sound will show less of a change in the contribution of sea ice-derived carbon in their diet compared to those residing to the south, in Foxe Basin where a more rapid decline in sea ice will occur. We also hypothesized that there will be no temporal change in $\delta^{15}N$ of the baseline and the trophic position of Atlantic walrus in both Jones Sound and Foxe Basin indicating consistent biogeochemical characteristics of nitrogen source availability in these areas and similar trophic roles of primary consumer consumption over time.

## 2. Material and methods

### (a) Environmental data
Summer sea ice concentrations of Jones Sound and Foxe Basin were obtained using Canadian Ice Service's IceGraph 2.0 tool (http://iceweb1.cis.ec.gc.ca/IceGraph, accessed April 2020) which disseminates sea ice data on regional and sub-regional spatial scales across the Canadian Arctic. We queried databases for each of the pre-defined sub-regions of Jones Sound and Foxe Basin, which encompassed the polynyas where walrus overwinter, for total accumulated ice concentrations from early July to end of August per year from 1982 to 2016. Sea ice breakup date was determined from the ordinal date on which the weekly sea ice concentration reached and remained below 50% in Jones Sound and Foxe Basin.

### (b) Sample collection
Paired Atlantic walrus liver and muscle samples were collected opportunistically from the northern region (Grise Fiord, Nunavut ($n = 10$)) and the southern region (Hall Beach, Nunavut ($n = 4$) and Igloolik, Nunavut ($n = 34$)) during July and August, 1982–2016, by Inuit hunters as part of their subsistence harvests (see figure 1 and table 1 for $n$ per sampling year). Hall Beach and Igloolik are only 70 km apart (figure 1a) and Atlantic walrus samples collected from these nearby communities were grouped together because they are part of the same management unit in northern Foxe Basin [20]. Atlantic walruses from Grise Fiord (1996 and 2006) are part of the North Water management unit of the high-Arctic population which do not overlap with the Foxe Basin management unit [18]. Tissues were stored at −20°C before processing. Based on average body mass of adult Atlantic walruses (approx. 1000 kg), the stable isotope half-life of muscle represents approximately six months which encompasses part of their overwintering period in nearby polynyas, as well as spring and summer foraging periods [20]. We performed highly branched isoprenoid diatom lipid biomarker analysis on Atlantic walrus liver samples, as

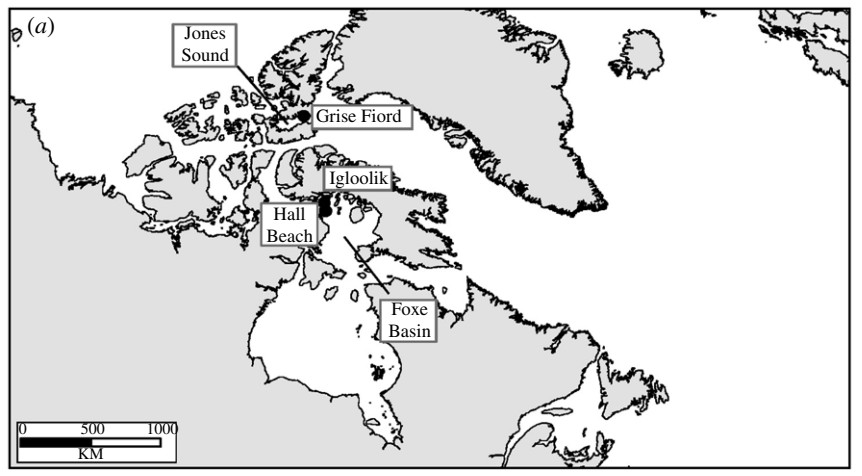

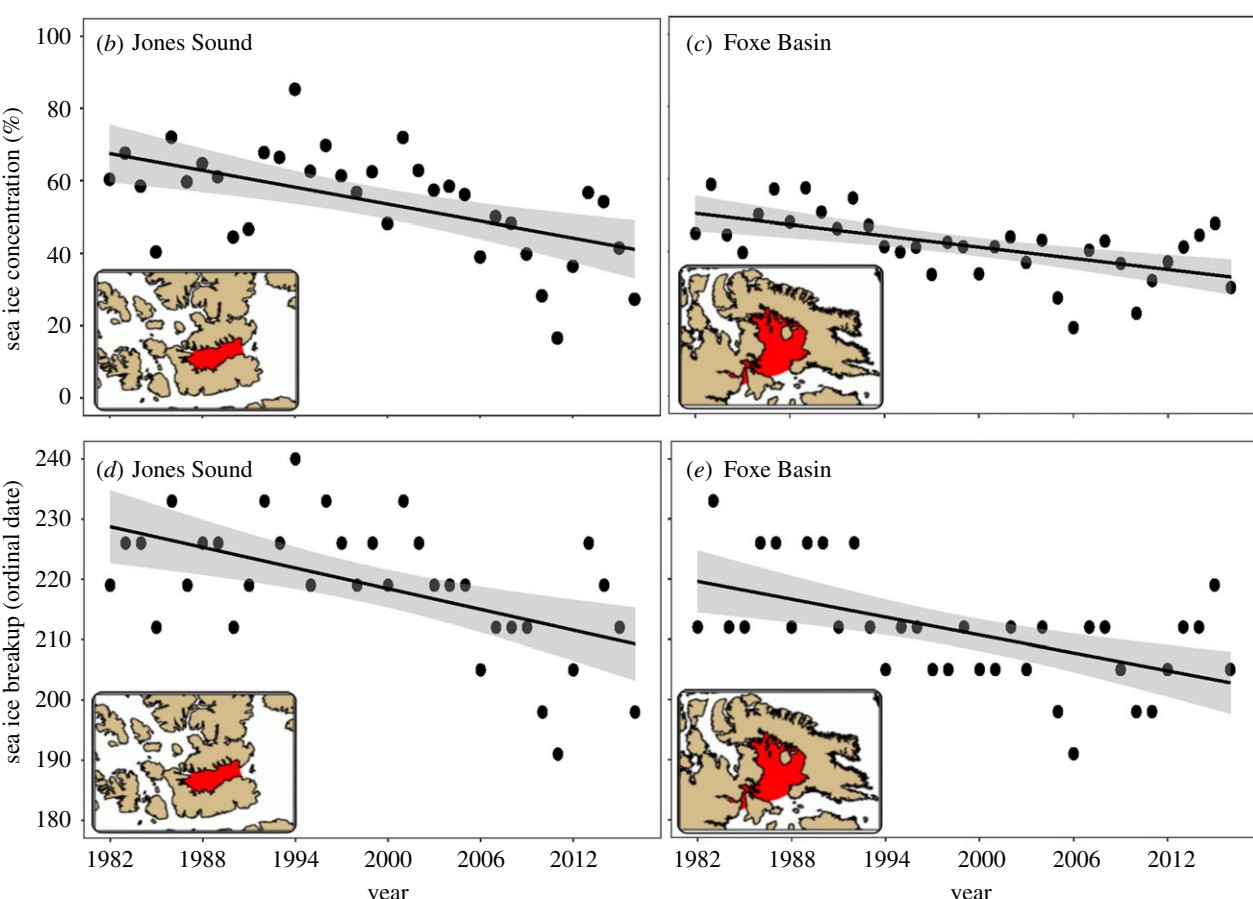

**Figure 1.** Atlantic walrus sample locations of Grise Fiord, Igloolik and Hall Beach (*a*) as well as July–August sea ice concentration and sea ice breakup date in Jones Sound (*b,d*) and Foxe Basin (*c,e*) from 1982 to 2016. Map insets for (*b*) and (*c*) were obtained from IceGraph 2.0 (Canadian Ice Service) highlighting the area where sea ice concentrations were estimated. July–August sea ice concentration significantly declined over time in Jones Sound (slope = −0.008, $t_{33} = -3.81$, $r^2 = 0.31$, $p < 0.001$) and in Foxe Basin (slope = −0.005, $t_{33} = -4.13$, $r^2 = 0.34$, $p < 0.001$). In addition, sea ice breakup date significantly decreased over time in Jones Sound (slope = −0.57, $t_{33} = -3.86$, $r^2 = 0.29$, $p < 0.001$) and Foxe Basin (slope = −0.49, $t_{33} = -3.75$, $r^2 = 0.30$, $p < 0.001$). (Online version in colour.)

greater than 70% of highly branched isoprenoid diatom lipids are typically stored in vertebrate liver [32]. The residency time of highly branched isoprenoid lipids in liver of larger mammals is unknown but is thought to be days to weeks, a slightly longer timeframe to that of primary consumers (i.e. days [22,24]) yet still represents a timeframe to investigate short-term contributions. For example, Brown *et al.* [21] used highly branched isoprenoids of ringed seal (*Pusa hispida*) liver samples to quantify monthly changes in sea ice-derived carbon contributions to ringed seal diet. Using both liver and muscle samples of Atlantic walrus allows one to measure time-integrated resource use over two different temporal scales (i.e. monthly using liver and seasonal using muscle). Stomach contents of Atlantic walrus (*n* = 22) from

Igloolik in 1996 were mainly composed of bivalve siphons along with some remnants of sea cucumber remains (P.J.B. 2019, unpublished data) representing a benthivorous invertebrate diet consistent with previous walrus diet assessments from this region (July 1987 and 1988 [15]).

## (c) Lipid extraction, analysis and quantification

Highly branched isoprenoids were extracted from liver tissue using established techniques of Brown *et al.* [33] and Belt *et al.* [34] at the Scottish Association of Marine Sciences. Briefly, lyophilized liver subsamples (0.1–2 g) were saponified (approx. 5 ml $H_2O$ : MeOH, 1 : 9; 20% KOH; 60 mins, 70°C) and mixed

**Table 1.** Annual sample sizes ($n$), July–August sea ice concentration and mean ± s.d. for $\delta^{13}C$ and $\delta^{15}N$ of both phenylalanine ($\delta^{13}C_{Phe}$ and $\delta^{15}N_{Phe}$) and glutamic acid ($\delta^{13}C_{Glu}$ and $\delta^{15}N_{Glu}$) from muscle tissue, sea ice-derived carbon (i.e. H-print) from liver tissue and trophic position estimates derived from published equations of Atlantic walrus muscle tissue collected from 1996 and 2006 in Jones Sound, Nunavut, Canada and from 1982 to 2016 in Foxe Basin, Nunavut, Canada.

| year | $n$ | sea ice concentration (%) | sea ice breakup (ordinal date) | $\delta^{13}C_{Phe}$ (‰) | $\delta^{13}C_{Glu}$ (‰) | sea ice-derived carbon (%) | $\delta^{15}N_{Phe}$ (‰) | $\delta^{15}N_{Glu}$ (‰) | trophic position[a] | trophic position[b] | trophic position[c] |
|---|---|---|---|---|---|---|---|---|---|---|---|
| Jones Sound | | | | | | | | | | | |
| 1996 | 5 | 70 | 233 | −25.2 ± 0.4 | −16.6 ± 1.8 | 97.9 ± 2.0 | 6.4 ± 1.1 | 19.1 ± 0.7 | 2.2 ± 0.2 | 3.1 ± 0.2 | 2.9 ± 0.2 |
| 2006 | 5 | 40 | 205 | −26.0 ± 0.3 | −17.6 ± 0.6 | 88.5 ± 15.5 | 6.1 ± 0.5 | 17.9 ± 0.8 | 2.1 ± 0.1 | 3.0 ± 0.1 | 2.8 ± 0.1 |
| Foxe Basin | | | | | | | | | | | |
| 1982 | 5 | 45 | 212 | −26.6 ± 0.4 | −16.5 ± 0.5 | 82.5 ± 10.8 | — | — | — | — | — |
| 1983 | 5 | 58 | 233 | −26.1 ± 0.6 | −17.2 ± 2.2 | 93.9 ± 3.8 | 9.2 ± 0.6 | 20.1 ± 1.4 | 2.0 ± 0.2 | 2.9 ± 0.2 | 2.6 ± 0.2 |
| 1987 | 4 | 57 | 226 | −27.2 ± 0.5 | −17.8 ± 0.8 | 94.4 ± 4.8 | — | — | — | — | — |
| 1988 | 5 | 48 | 212 | −27.2 ± 0.5 | −19.5 ± 2.3 | 81.7 ± 6.3 | 8.3 ± 0.9 | 18.7 ± 0.4 | 1.9 ± 0.1 | 2.8 ± 0.1 | 2.5 ± 0.1 |
| 1996 | 5 | 41 | 212 | −27.8 ± 0.8 | −18.4 ± 0.8 | 49.1 ± 30.7 | 6.6 ± 1.6 | 19.1 ± 0.5 | 2.2 ± 0.2 | 3.1 ± 0.2 | 2.9 ± 0.2 |
| 2008 | 5 | 43 | 212 | −27.3 ± 0.4 | −19.1 ± 0.7 | 66.0 ± 19.7 | 9.0 ± 0.3 | 19.1 ± 1.1 | 1.9 ± 0.1 | 2.8 ± 0.1 | 2.5 ± 0.1 |
| 2009 | 5 | 37 | 205 | −28.7 ± 0.7 | −19.6 ± 1.6 | 41.7 ± 10.9 | 9.0 ± 0.4 | 20.1 ± 0.6 | 2.0 ± 0.1 | 2.9 ± 0.1 | 2.7 ± 0.1 |
| 2013 | 2 | 41 | 212 | −28.0 ± 0.1 | −18.2 ± 0.5 | 4.1 ± 3.1 | — | — | — | — | — |
| 2016 | 2 | 30 | 205 | −28.1 ± 0.2 | −17.8 ± 0.7 | 32.5 ± 3.7 | 7.8 ± 2.3 | 20.8 ± 3.0 | 2.3 ± 0.1 | 3.2 ± 0.1 | 3.0 ± 0.1 |

[a]Chikaraishi et al. [28].
[b]Germain et al. [30].
[c]McMahon et al. [31].

with hexane ($3 \times 4$ ml), then centrifuged (2 min; 2500 revolutions per minute) with hexane prior to being transferred to clean glass vials and then dried with $N_2$ stream to remove traces of $H_2O$ and MeOH. Dried lipid extracts were then fractionated (5 ml of hexane) using column chromatography ($SiO_2$; 0.5 g). Highly branched isoprenoids were analysed by gas chromatography mass spectrometry and quantified by measuring the mass spectral intensities for each HBI in selective ion monitoring mode (see [23]). Percentages of sea ice-derived carbon were quantified from H-Print estimates using analytical intensities of three highly branched isoprenoids ($IP25$: $m/z$ 350.3, II; $m/z$ 348.3 and III; $m/z$ 346.3) according to equation (2.1), with this combination allowing a linear calibration to be constructed [33]

$$H - Print(\%) = \frac{(III)}{(IP_{25} + II + III)} \times 100. \quad (2.1)$$

Sea ice-derived carbon, as a proportion of marine origin carbon within samples, was estimated using equation (2) from previous H-Print calibration ($R^2 = 0.97$, $p \leq 0.01$, d.f. = 23) [33] and expressed below as mean values.

$$sea \ ice - derived \ carbon \,(\%) = 101.08 - 1.02 \times H - print. \quad (2.2)$$

## (d) Compound-specific stable isotope analysis

The $\delta^{13}C$ and $\delta^{15}N$ values of individual amino acids were analysed via gas chromatography-combustion isotope ratio mass spectrometry at University of California, Davis where approximately 3 mg of lyophilized and homogenized Atlantic walrus muscle samples underwent acid hydrolysis 6 M HCl at 150°C under a $N_2$ headspace for 70 min and derivatized using methoxycarbonylation esterification (see [35,36] for more details). Quality assurance of $\delta^{13}C$ and $\delta^{15}N$ followed protocols of [36] where two mixtures composed of pure amino acids of calibrated $\delta^{13}C$ and $\delta^{15}N$ (UCD AA1, UCD AA2) were co-measured with samples. In addition, two well-described, natural materials were co-analysed with samples and used as secondary quality assurance materials (baleen and fish muscle). We measured the $\delta^{13}C$ and $\delta^{15}N$ values from a total of 12 amino acids but focus on two that are most commonly used as a trophic and baseline amino acid: glutamic acid (Glu), a non-essential, trophic amino acid; and phenylalanine (Phe), an essential, source amino acid. Mean analytical precision assessed from duplicate measures of internal baleen and fish muscle samples and two internal reference compounds (UCD AA1 and UCD AA3) were less than or equal to 0.3‰ for $\delta^{13}C$ and less than or equal to 1‰ for $\delta^{15}N$. All stable isotope ratios are expressed in per mil (‰) in standard delta (δ) notation relative to the international standards Pee Dee Belemnite for carbon and atmospheric $N_2$ for nitrogen, using the following equation: $\delta X = [(Rsample/Rstandard) - 1] \times 10^{-3}$, where X is $^{13}C$ or $^{15}N$ and R equals $^{13}C/^{12}C$ or $^{15}N/^{14}N$.

## (e) Statistical analysis

We used linear regression analysis to investigate the relationship between $\delta^{13}C$ of phenylalanine ($\delta^{13}C_{Phe}$) and sea ice-derived carbon from H-print estimates—two variables that have been used to discriminate between sea ice-derived carbon and phytoplankton-derived carbon [32,37]. A lower $\delta^{13}C_{Phe}$ value represents a higher contribution of phytoplankton-derived carbon to the diet of the consumer. A Pearson correlation was used to examine the correlation between summer sea ice concentration and sea ice breakup date. We also used linear regression analysis to investigate temporal trends of sea ice concentration in Jones Sound and Foxe Basin. To test for changes across years in Foxe Basin, mean annual values of $\delta^{13}C_{Phe}$ (i.e. baseline $\delta^{13}C$), $\delta^{15}N_{Phe}$ (i.e. baseline $\delta^{15}N$) and sea ice-derived carbon (i.e. derived from H-print values) were used in linear regressions to meet the

assumption of independent data for this time series (1982–2016; $n = 9$ for $\delta^{13}C_{Phe}$ and sea ice-derived carbon, and $n = 6$ for $\delta^{15}N_{Phe}$) since one value of sea ice concentration was estimated for each year and generally five Atlantic walrus individuals were collected per year, except in 1987 ($n = 4$), 2013 ($n = 2$) and 2016 ($n = 2$). Including each individual within each sampling year violates the assumption of independent errors and allows pseudoreplication where observations from the same year are more likely to be similar than observations from other years. We used a similar analytical approach to investigate relationships of $\delta^{13}C_{Phe}$ and sea ice-derived carbon relative to sea ice concentration of Atlantic walruses in Foxe Basin. Student $t$-tests were used to determine a significant difference in $\delta^{13}C_{Phe}$, $\delta^{15}N_{Phe}$ and sea ice-derived carbon between 1996 versus 2006 for individuals collected in Jones Sound.

There are several trophic position (TP) equations derived from compound-specific stable isotope analysis of individual amino acids that provide a range in trophic position estimates for marine mammals and seabirds [38,39]. Given the high amount of variation in trophic discrimination factors between glutamic acid and phenylalanine ($TDF_{Glu-Phe}$) related to a species' mode of nitrogen excretion and diet quality [27], multi-TDF equations are considered more accurate and can improve trophic position estimates of consumers compared to single, step-wise TDF equations [30,31,39], and provide further support for a scaled TDF and trophic level framework [40]. Therefore, we estimated Atlantic walrus trophic position using a multi-TDF approach that incorporated a seal-specific TDF [30])

$$TP = 2 + \left\{ \frac{\delta^{15}N_{Glu} - \delta^{15}N_{Phe} - \ TDF_{Glu-Phe}}{\beta} \right\}, \quad (2.3)$$

where $\delta^{15}N_{Glu}$ and $\delta^{15}N_{Phe}$ represent the stable isotope nitrogen values of glutamic acid and phenylalanine of the consumer, $\beta$ represents the difference in $\delta^{15}N_{Glu}$ and $\delta^{15}N_{Phe}$ of primary producers (3.4‰) [25] and $TDF_{Glu-Phe}$ is the seal-specific TDF value of 4.3‰. Alternative methods for calculated trophic position of Atlantic walrus provided differences in absolute values but similar trends over time (see electronic supplementary material, S1). To test for temporal changes in trophic position of Atlantic walruses in Foxe Basin, linear regression was used on mean annual values of trophic position over time. A student $t$-test was used to determine a significant difference in trophic position of Jones Sound Atlantic walrus between 1996 and 2006. All statistical analyses were performed in R v. 3.6.1 [41].

## 3. Results

### (a) Temporal trends in sea ice concentration

Jones Sound had a consistently higher summer sea ice concentration than in Foxe Basin by 13% ± 12% (average ± s.d.) each year from 1982 to 2016. However, over this 35-year record, summer sea ice concentration decreased by approximately 26% in Jones Sound (slope = −0.78, intercept = 1607.7, $t_{33} = −3.81$, $r^2 = 0.31$, $p < 0.001$) and 18% in Foxe Basin (slope = −0.52, intercept = 1071.8, $t_{33} = −4.13$, $r^2 = 0.34$, $p < 0.001$; figure 1). In addition, sea ice breakup date now occurs 20 days earlier in Jones Sound (slope = −0.57, intercept = 1363.50, $t_{33} = −3.68$, $p < 0.001$) and 17 days earlier in Foxe Basin (slope = −0.49, intercept = 1198.90, $t_{33} = −3.75$, $p < 0.001$; figure 1). Sea ice breakup date was highly correlated with summer sea ice concentration (0.92) in that an early sea ice breakup date was associated with a lower summer sea ice concentration and a later sea ice breakup was correlated with higher summer sea ice concentrations. Therefore,

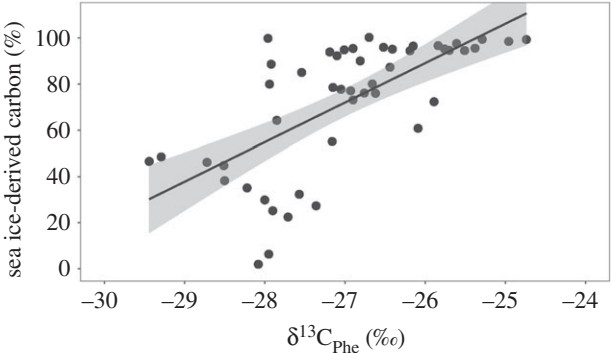

**Figure 2.** Linear regression (black line) with 95% confidence intervals (grey) of $\delta^{13}C_{Phe}$ relative to sea ice-derived carbon (slope = 17.12, $t_{46}$ = 6.11, $r^2$ = 0.45, $p < 0.001$) of Atlantic walrus tissues from 1982 to 2016.

sea ice breakup date was not included in linear models due to multicollinearity.

### (b) Relationship between sea ice-derived carbon and $\delta^{13}C_{Phe}$

Sea ice-derived carbon was strongly positively correlated with $\delta^{13}C_{Phe}$ ($r = 0.67$). Furthermore, a significant increase in $\delta^{13}C_{Phe}$ relative to sea ice-derived carbon occurred (slope = 17.12, intercept = 534.32, $t_{46} = 6.11$, $r^2 = 0.45$, $p < 0.001$) with estimated sea ice-derived carbon values ranging from 30% to greater than 100% in association with $\delta^{13}C_{Phe}$ values ranging from −29.4‰ to −24.7‰ (table 1 and figure 2). Given the moderate explanatory power of this linear equation, future studies can then use this model to estimate the percentage of sea ice-derived carbon in the diet of Arctic species using $\delta^{13}C_{Phe}$ values from their study species.

### (c) Temporal trends of sea ice-derived carbon, $\delta^{13}C_{Phe}$, $\delta^{15}N_{Phe}$ and trophic position

In Jones Sound, there were no significant differences in sea ice-derived carbon ($t_8 = 1.35$, $p = 0.25$), $\delta^{15}N_{Phe}$ ($t_8 = 0.57$, $p = 0.59$) and trophic position ($t_8 = 1.01$, $p = 0.34$) of Atlantic walrus between 1996 and 2006 (table 1). However, $\delta^{13}C_{Phe}$ was 0.8‰ lower in 2006 versus 1996 ($t_8 = 3.81$, $p < 0.01$) when sea ice concentration was 70% in 1996 and only 40% in 2006 (table 1). In Foxe Basin, $\delta^{15}N_{Phe}$ ($t_5 = 0.19$, $p = 0.85$) and trophic position ($t_5 = 0.63$, $p = 0.53$) of Atlantic walrus did not change over time, whereas significant decreases occurred for sea ice-derived carbon ($t_8 = -4.60$, $p < 0.01$) and $\delta^{13}C_{Phe}$ ($t_8 = -3.80$, $p < 0.01$; figure 3). Both sea ice-derived carbon ($t_8 = 3.30$, $p = 0.01$) and $\delta^{13}C_{Phe}$ ($t_8 = 3.20$, $p < 0.02$) were significantly higher in years of higher sea ice concentration (figure 4).

## 4. Discussion

Using a novel biochemical coupling approach for polar environments, we identified a prominent 35-year decline (i.e. 1982–2016) in the contribution of sea ice-derived carbon to Atlantic walrus, a benthic fauna consumer, in Foxe Basin. This pattern was associated with a dramatic decline in sea ice cover where Atlantic walrus shifted from a nearly exclusive contribution of sea ice-derived carbon in their diet to more

phytoplankton-derived carbon. In Jones Sound, a slight decrease of sea ice-derived carbon to the benthic environment occurred between 1996 versus 2006, though Atlantic walrus in this area have maintained a nearly exclusive amount of sea ice-derived carbon in their diet (98% in 1996 and 89% in 2006) despite a large difference in summer sea ice concentration between sampling years (70% in 1996 versus 40% in 2006). We found strong support of a decoupling of sea ice–benthic habitats. Less sea ice-derived carbon now reaches benthic fauna at mid-Arctic, but not high-Arctic, latitudes which suggests spatial variation in fundamental alterations of carbon energy dynamics and ecosystem functioning due to climate change. This result aligns with observed changes in sea ice algal bloom phenology and production in that the sea ice algal bloom is occurring earlier in the year across the Arctic, but there is latitudinal heterogeneity in the amount of sea ice algal production [42]. In the near future, high-Arctic latitudes (74° N to 84° N; e.g. Jones Sound) may experience an 11–550% increase in ice algal production due to a shifting of the ice season to more favourable photoperiods, whereas a smaller change or loss (−25% to 73%) in ice algal production will occur at mid-Arctic latitudes (66° N to 74° N; e.g. Foxe Basin) due to a narrowing of the sea ice algal production window [42]. This finding is further strengthened by evidence of a consistent stable nitrogen isotope baseline over the same time period, indicating that the biogeochemical characteristics in terms of $\delta^{15}N$ of seawater nitrate and nitrogen cycling dynamics of Jones Sound and Foxe Basin have remained the same between 1996–2006 and 1982–2016, respectively. In addition, Atlantic walruses occupied a steady trophic position of approximately 3 in both areas, suggesting consistent trophic roles of consuming filter-feeding bivalves and other benthic primary consumers.

### (a) Spatio-temporal variation of sea ice-derived carbon to benthic fauna

Sea ice-derived carbon estimated from highly branched isoprenoids has been observed across all consumer trophic levels from primary and secondary consumers to near-top and top predators [21–24,43], suggesting at least some dependency and continued use of sea ice-derived carbon by Arctic biota throughout the year, particularly in the benthic environment [11,43–45]. However, inter-annual variability in the availability of sea ice algae to consumers in the ecosystem and in turn the strength of sympagic-pelagic-benthic coupling is contemporaneous with inter-annual changes in environmental conditions [45]. Over a 9-year period from 2002–2010 in the Bering Sea, sea ice algae carbon was less available to consumers such as ice seals and their prey, in both pelagic and benthic environments in warmer years with less sea ice than in relatively colder years with more sea ice [45]. In addition, the amount of sea ice-derived carbon in the diet of beluga whales (*Delphinapterus leucas*) and ringed seals (*Pusa hispida*), predators who mainly consume prey within the epi-pelagic portion of the water column, has declined over a 27-year study period in Cumberland Sound, Nunavut [21,23]. Using Atlantic walrus as a bioindicator for carbon cycling in benthic communities, this study provides the first empirical support for a multi-decadal decline (1980–2010s) of sea ice-derived carbon in the benthic environment in association with sea ice phenology and cover that varies with latitude.

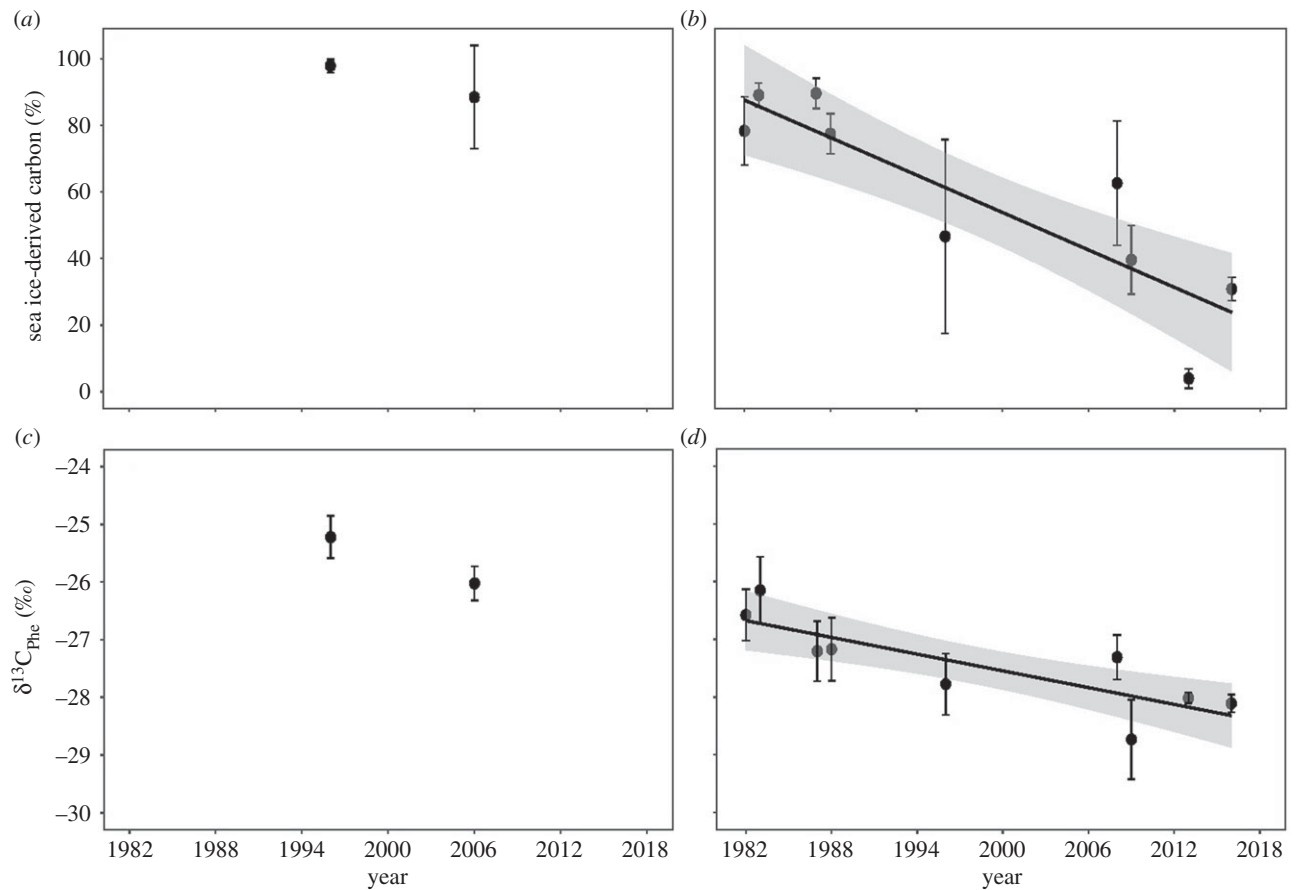

**Figure 3.** Temporal trends of sea ice-derived carbon and $\delta^{13}C_{Phe}$ of Atlantic walrus from Jones Sound (a,c) and Foxe Basin (b,d). Linear regressions (black line) with 95% confidence intervals (grey) were performed on mean annual values. Error bars represent standard deviation. Sea ice-derived carbon (slope = −1.98, $t_7$ = −4.60, $r^2$ = 0.75, $p$ = 0.002) and $\delta^{13}C_{Phe}$ (slope = −0.05, $t_7$ = −3.80, $r^2$ = 0.67, $p$ = 0.006) for Atlantic walruses in Foxe Basin significantly declined over time.

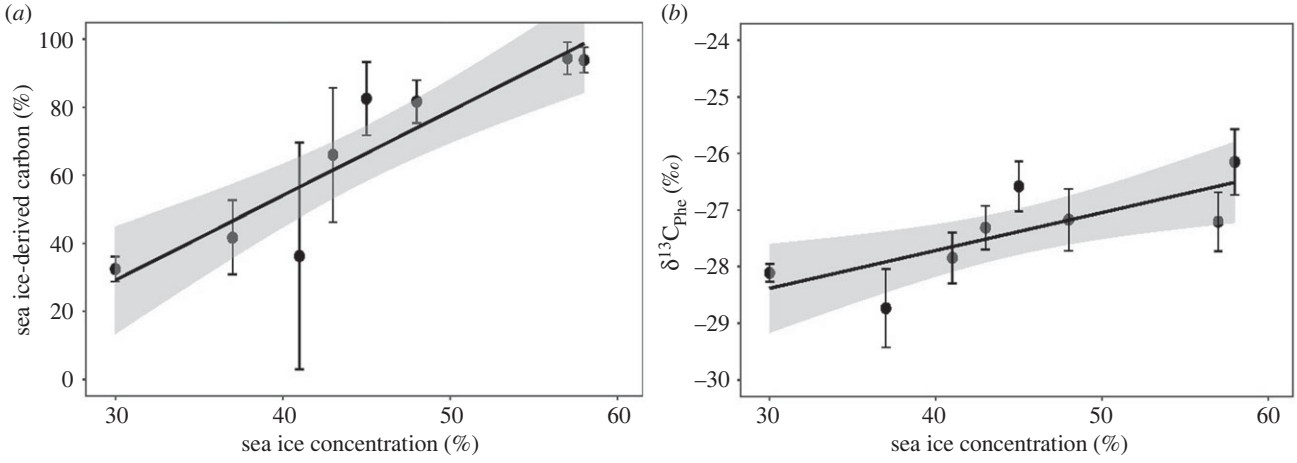

**Figure 4.** Linear regressions (black line) with 95% confidence intervals (grey) of sea ice-derived carbon (a) and $\delta^{13}C_{Phe}$ (b) of Foxe Basin Atlantic walrus relative to July–August sea ice concentration. Mean annual values were used for analysis. Error bars represent standard deviation. Both sea ice-derived carbon (slope = 2.69, $t_7$ = 3.31, $r^2$ = 0.61, $p$ = 0.01) and $\delta^{13}C_{Phe}$ (slope = 0.07, $t_7$ = 3.17, $r^2$ = 0.59, $p$ < 0.02) were significantly higher relative to increasing sea ice concentration.

In the 1980s in the more-southern area of Foxe Basin, the contribution of sea ice-derived carbon (i.e. highly branched isoprenoid diatom lipids from liver) in the diet of Atlantic walrus was similar to that of the more-northern area of Jones Sound in the 1990s and 2000s (88% versus 94%), but then decreased dramatically in the 1990s and 2000s where sea ice-derived carbon became 1.8 times lower (52%) than in Jones Sound. More rapid declines in summer sea ice concentration and earlier sea ice breakup in Jones Sound relative to Foxe Basin from 1982 to 2016 (this study) is consistent with observations of greater

susceptibility of northern Arctic regions to reduced springtime ice seasons than more-southern areas [46]. Despite this, the considerable decrease in the contribution of sea ice-derived carbon observed in Foxe Basin Atlantic walrus over time suggests a much greater reduction of sea ice algal production and transportation to the benthos at lower latitudes than at higher latitudes, or less ice algae-dependent prey being available. This decrease may represent a direct negative impact on benthic biomass and community structure in Foxe Basin as sea ice algae provides nutritionally rich lipids and relatively undegraded carbon to the

seafloor in the spring, which in turn negatively impacts the functioning of the lipid-reliant Arctic ecosystem [3,43].

The Arctic seafloor and its sediments act as a long-term repository for sea ice-derived carbon where benthic heterotrophs feed on these sedimentary stores of sea ice-derived carbon which can then be transferred back up to the pelagic environment via fishes, marine mammals and seabirds who consume benthic fauna [43]. Therefore, the spatio-temporal differences of sea ice-derived carbon in the benthos suggests that Jones Sound, a higher latitude area with a longer ice-covered period, could harness greater sea ice-derived carbon deposits in the benthos at present than in Foxe Basin, potentially buffering the system against sea ice loss. However, whether this potential carbon buffering in Jones Sound continues into the future with even less sea ice, or whether phytoplankton-derived carbon will mainly fuel this benthic system in the future requires further monitoring as results observed from Foxe Basin may foreshadow that of Jones Sound.

The $\delta^{13}C_{Phe}$ of Atlantic walrus in Foxe Basin declined at a rate of 0.6‰ per decade from 1982 to 2016 in association with a decreasing sea ice concentration and earlier sea ice breakup, and was 0.8‰ lower in 2006 than in 1996 in Jones Sound, suggesting less sea ice-derived carbon in the benthos over time in both areas which aligns with our H-Print estimates. The decadal declines of $\delta^{13}C_{Phe}$ in Atlantic walrus from both study areas is six to eight times higher than the decrease of $\delta^{13}C$ of dissolved inorganic carbon in Arctic marine waters (−0.11‰ per decade) attributed to increasing concentrations of anthropogenic $CO_2$ in the atmosphere known as the Suess effect [47]. Additionally, our decadal declines in $\delta^{13}C_{Phe}$ parallel estimates from [47] where the entire pool of $\delta^{13}C$ of particulate organic carbon in the Arctic Ocean decreased by 0.6‰ per decade with declining sea ice in combination with increased phytoplankton primary productivity [5]. It is unlikely that bacterial and meiofaunal processes in the benthic substrate contributed to higher $\delta^{13}C_{Phe}$ values over time [48] since temporal decreases in $\delta^{13}C_{Phe}$ corresponded to that of temporal declines in H-Print values. From 1998 to 2018, both Jones Sound and Foxe Basin have undergone an increase in phytoplankton biomass [5] which in turn, sinks as phytodetritus to then be exploited by benthic consumers. Net primary productivity across the Arctic has increased by 57% from 1998 to 2018 due to increased phytoplankton biomass supported by an influx of new nutrient availability [5], however, it is unknown whether this ever-increasing phytoplankton production can subsidize the continual loss of sea ice algae for benthic consumers. Arctic bivalve growth rate is greater in years with more sea ice [49] due to the availability of more nutritionally rich polyunsaturated lipids from sea ice algae than from phytoplankton [50,51]. Therefore, reductions in sea ice cover and amount of sea ice algae that reaches the benthos may negatively influence the growth rate and body size of Arctic bivalve communities and in turn, benthic foragers, such as Atlantic walrus. In addition, it is unknown whether a corresponding reorganization of benthic community structure in Foxe Basin has also occurred.

## (b) Spatio-temporal variation of Atlantic walrus $\delta^{15}N_{Phe}$ and trophic position

Nitrogen is a key element of life and is one of the major elements required for primary production in marine environments [52]. Nitrogen isotopes in marine primary producers

can be used to trace the biogeochemical cycling of oceanic processes as primary producers assimilate nitrate, therefore their $\delta^{15}N$ reflects the $\delta^{15}N$ of seawater nitrate [52]. Both Jones Sound and Foxe Basin are influenced by Pacific waters which are first modified by physical and biogeochemical processes in the Arctic Basin prior to entering and exiting the Canadian Archipelago eastward through Lancaster Sound and southward via Fury and Hecla Straits [53,54]. Though the inflow of Pacific water has increased to the Arctic from 1990 to 2015 [55], no long-term changes to the $\delta^{15}N$ of the baseline in Jones Sound and Foxe Basin occurred suggesting that water circulation, nitrogen cycling and nitrate assimilation by primary producers from both areas have remained consistent over time.

Trophic position estimates revealed that Atlantic walrus in Jones Sound and Foxe Basin have consistently played the same trophic role in each area over time. Equation (2.3) provided a trophic position of Atlantic walrus (range = 2.8–3.2) that is in better agreement with estimates derived from stomach contents (3.4) [56] compared to equations that use a universal trophic discrimination of 7.6‰ [28] which generally underestimate the trophic position of marine mammals [38]. Though Atlantic walrus have been documented to occasionally consume ringed seals [16] who occupy a trophic position of 3.3 to 4.6 [57], their contribution to Atlantic walrus diet is very low, consistent with our trophic position estimates of approximately 3. Despite declining sea ice concentration and Atlantic walrus habitat in both Jones Sound and Foxe Basin, both management units have had stable population estimates since the 1970s, suggesting healthy populations over the duration of our study period [58].

## (c) Summary

Our novel biochemical trophodynamic approach using compound-specific stable isotope analysis of amino acids in combination with highly branched isoprenoids revealed that: (1) both analytical approaches are complementary to each other with moderate explanatory power, (2) in association with an earlier sea ice breakup and declining sea ice concentration, a strong decoupling of sea ice-benthic habitats occurred at mid-Arctic latitudes where less sea ice algae is now incorporated into benthic fauna, (3) sea ice algae is the nearly exclusive carbon source for benthic fauna in the high-Arctic and (4) the $\delta^{15}N$ of the baseline and trophic position of Atlantic walrus did not change over time indicating consistent nitrogen cycling and trophic roles of high bivalve consumption. We identified spatio-temporal changes in the carbon source that fuels benthic communities in the Arctic, suggesting a continuing decrease and thinning of sea ice cover will influence species interactions, benthic community biomass and structure, and Arctic ecosystem functioning. Long-term monitoring of Atlantic walrus biological parameters, such as body condition, is required to determine how Atlantic walrus and potentially other benthic consumers are responding to these changing carbon sources to gain insight into the modifications in ecosystem dynamics across the rapidly warming Arctic.

Data accessibility. Data are available from the Dryad Digital Repository: https://doi.org/10.5061/dryad.12jm63xwj [59].
Authors' contributions. D.J.Y. conceived and designed the study, conducted the analysis and drafted the manuscript. P.J.B. and S.H.F. provided samples and data, and T.A.B. performed the highly

branched isoprenoid analysis. All authors contributed input and approved the final manuscript version.

Competing interests. We declare no competing interests.

Funding. This study was supported by Fisheries and Oceans Canada, Nunavut Implementation Fund, Marine Environmental Observation Prediction and Response Network, Natural Sciences and Engineering Research Council of Canada and the W. Garfield Weston Foundation.

Acknowledgements. We thank the Igloolik, Hall Beach and Grise Fiord Hunters and Trappers Organizations and their hunters for collecting Atlantic walrus samples. We thank the two reviewers and the associate editor for thoughtful comments that improved this manuscript. We thank Krista Kenyon for sample processing and Chris Yarnes and the UC Davis Stable Isotope Facility for compound-specific stable isotope analysis of amino acids. This article is dedicated to Lyla Jane Yurkowski.

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
