## [Reviewer comments · Proceedings of the Royal Society B: Biological Sciences]

Review History

RSPB-2020-2126.R0 (Original submission)

Review form: Reviewer 1

Recommendation

Accept with minor revision (please list in comments)

Scientific importance: Is the manuscript an original and important contribution to its field?

Excellent

General interest: Is the paper of sufficient general interest?

Good

Quality of the paper: Is the overall quality of the paper suitable?

Excellent

Is the length of the paper justified?

Yes

Should the paper be seen by a specialist statistical reviewer?

No

Do you have any concerns about statistical analyses in this paper? If so, please specify them explicitly in your report.

No

It is a condition of publication that authors make their supporting data, code and materials available - either as supplementary material or hosted in an external repository. Please rate, if applicable, the supporting data on the following criteria.

Is it accessible?

Yes

Is it clear?

Yes

Is it adequate?

Yes

Do you have any ethical concerns with this paper?

No

Comments to the Author

This is a very nice study with some surprising results, and a very well written paper! I only have some relatively minor suggestions, and I'm looking forward to see the paper published.

Title

Would 'Atlantic walrus signal a decrease in sea ice-derived carbon to benthic fauna in the mid-latitude Canadian Arctic' be more appropriate? From your results, you found a strong decline in the utilization of sea ice-derived carbon in mid-latitude walrus, but it remained similarly high in Jones Sound. You should also consider including something like 'long-term' to highlight the strength of your dataset and this study.

Keywords: add Amino acids or replace CSIA with Amino acid-specific SIA

Abstract

L26: Consider replacing 'integrate' with 'combine' or 'coupled'.

LL 30-32: I'm not sure I understand the sentence. Wouldn't it suggest a 'coupling' instead of 'decoupling' when the contribution of sea-ice derived carbon decreases with decreasing sea ice?

L36: Maybe instead of 'benthic fauna', you can say 'Atlantic walrus' since you didn't investigate different benthic fauna in this study, but walrus that is preying on this fauna.

How are your results related to 'a change in Arctic marine ecosystem functioning'? Maybe you can be a bit more specific (or leave it out)?

Specify in the Abstract that you studied samples from the Canadian Arctic, e.g. L 30 '... at high- and mid-latitudes in the Canadian Arctic'.

Introduction

The Introduction is well written, but a bit lengthy, could be condensed. Consider e.g. shortening the first two paragraphs about primary production and climate change, for example LL58-60 could be skipped, and maybe also the following sentence about the mismatch for grazers as it is not really related to your study, and LL82-83 as you later on in the Methods section state that samples were obtained by Inuit harvesting, and/or LL71-89.

L53: '...drives the energy flow...'

L100: Consider adding phenylalanine in brackets for the essential AAs and also add the term 'source AA' as this comes up later on in the text.

L102: I'm not sure why you compare the AA-CSIA to BSIA since you didn't run bulk analysis.

Maybe you only focus on the methods you actually applied, and you could skip LL102-106 (also to shorten the text).

L107: Put glutamic acid in brackets after trophic AA.

L115: I'm not sure if generalizing as 'benthic fauna' is a good idea, maybe stick to 'Atlantic walrus'.

L118: 'Predicted' sounds like you run actual models to predict something. Maybe rephrase to 'we hypothesize'. The same further down in this paragraph.

LL124-127: But wouldn't it be possible/likely to see a change in $\delta^{15}\text{N}$ as availability, concentrations of N-sources change due to environmental changes?

Methods

This section is very well and clearly written.

L 152: Can you specify in which lab the HBI samples have been analysed?

LL197-199: You could add a couple of references for this statement.

L208: Table 1 shows that there were between 2 and 5 individuals per year, suggest to also specify in the text.

L219: I don't think the term TDF has been introduced yet. I saw it has been explained in the Supporting Information, but it should also be introduced here.

L232: Student's t-test? Paired/unpaired?

L234: Consider adding the packages you used in R.

Results

I wonder if it would be better to start the Results section with the basic environmental information about sea ice concentrations and temporal changes. So, you could consider to have (a) temporal trends in sea ice concentration including LL247-257, then (b) Relationship between sea ice-derived carbon and $\delta^{13}\text{CPhe}$ and (c) Temporal trends in sea ice-derived carbon, $\delta^{13}\text{CPhe}$, $\delta^{15}\text{NPhe}$ and trophic position.

Table 1: Is there a reason why you only present $\delta^{13}\text{CPhe}$ and $\delta^{13}\text{CGlu}$ from muscle tissue and % sea ice-derived carbon from liver tissue (the time frame of trophic information maybe?).

Consider stating in the Methods section why you used which tissue for which analysis.

L258: '...there were no significant differences..'

L258-266: You should more frequently refer to Table 1 where the actual values can be found.

Consider adding a sentence about the partly great differences in TP calculation using the different published equations. It has been explained well in the Supporting Information, but maybe you could state again that the higher estimations are likely the more realistic values.

Figure 1: Could you illustrate in the first map (A) where Foxe Basin and Jones Sound are located?

Discussion

This section is also very well written and explains the most important results of this study.

L292: You should probably say TP of 2-3 since there were large differences between the calculations, or put the range in brackets (1.9-3.2).

L318-L320: Could it also indicate a change in food sources (less ice algae-dependent prey)?

L311 and following: Can you specify from which tissue (muscle or liver) these values are derived from, so the reader can get some idea about time frames?

L322: You could probably be more specific to the season here, as ice algae during summer are likely less nutritional and attractive as a carbon source than earlier in the season.

L372-377: Here you could possibly elaborate a bit more on the differences in your TP calculations derived from the different equations. Why do you think the individuals in your study did not rely on seals? What could the stability of TP mean in terms of environmental changes, would walruses be rather resilient?

Review form: Reviewer 2

Recommendation

Accept with minor revision (please list in comments)

Scientific importance: Is the manuscript an original and important contribution to its field?

Good

General interest: Is the paper of sufficient general interest?

Good

Quality of the paper: Is the overall quality of the paper suitable?

Acceptable

Is the length of the paper justified?

Yes

Should the paper be seen by a specialist statistical reviewer?

No

Do you have any concerns about statistical analyses in this paper? If so, please specify them explicitly in your report.

No

It is a condition of publication that authors make their supporting data, code and materials available - either as supplementary material or hosted in an external repository. Please rate, if applicable, the supporting data on the following criteria.

Is it accessible?

Yes

Is it clear?

Yes

Is it adequate?

No

Do you have any ethical concerns with this paper?

No

Comments to the Author

See file attached. (See Appendix A)

Decision letter (RSPB-2020-2126.R0)

29-Oct-2020

Dear Dr Yurkowski:

Your manuscript has now been peer reviewed and the reviews have been assessed by an Associate Editor. The reviewers' comments (not including confidential comments to the Editor) and the comments from the Associate Editor are included at the end of this email for your

reference. As you will see, the reviewers and the Editors have raised some concerns with your manuscript and we would like to invite you to revise your manuscript to address them.

Research ethics:

Use of animals and field studies:

It is a condition of publication that you make available the data and research materials supporting the results in the article. Please see our Data Sharing Policies (<https://royalsociety.org/journals/authors/author-guidelines/#data>). Datasets should be deposited in an appropriate publicly available repository and details of the associated accession number, link or DOI to the datasets must be included in the Data Accessibility section of the article (<https://royalsociety.org/journals/ethics-policies/data-sharing-mining/>). Reference(s) to datasets should also be included in the reference list of the article with DOIs (where available).

Please submit a copy of your revised paper within three weeks. If we do not hear from you within this time your manuscript will be rejected. If you are unable to meet this deadline please let us know as soon as possible, as we may be able to grant a short extension.

Best wishes,
Dr Daniel Costa
mailto:proceedingsb@royalsociety.org

Associate Editor
Board Member: 1
Comments to Author:

Both reviewers assess the study positively, while including multiple suggestions for minor revision. In particular, as well as addressing the individual suggestions, the authors are advised to try and streamline the Introduction, and one reviewer in comments to Editor made the suggestion that data from individuals should be included to satisfy the 'data availability' criteria (pointing out that only mean data are included in the paper itself).

Reviewer(s)' Comments to Author:

Referee: 1

Comments to the Author(s)

This is a very nice study with some surprising results, and a very well written paper! I only have some relatively minor suggestions, and I'm looking forward to see the paper published.

Title

Would 'Atlantic walrus signal a decrease in sea ice-derived carbon to benthic fauna in the mid-latitude Canadian Arctic' be more appropriate? From your results, you found a strong decline in the utilization of sea ice-derived carbon in mid-latitude walrus, but it remained similarly high in Jones Sound. You should also consider including something like 'long-term' to highlight the strength of your dataset and this study.

Keywords: add Amino acids or replace CSIA with Amino acid-specific SIA

Abstract

L26: Consider replacing 'integrate' with 'combine' or 'coupled'.

LL 30-32: I'm not sure I understand the sentence. Wouldn't it suggest a 'coupling' instead of 'decoupling' when the contribution of sea-ice derived carbon decreases with decreasing sea ice?

L36: Maybe instead of 'benthic fauna', you can say 'Atlantic walrus' since you didn't investigate different benthic fauna in this study, but walrus that is preying on this fauna.

How are your results related to 'a change in Arctic marine ecosystem functioning'? Maybe you can be a bit more specific (or leave it out)?

Specify in the Abstract that you studied samples from the Canadian Arctic, e.g. L 30 '... at high- and mid-latitudes in the Canadian Arctic'.

Introduction

The Introduction is well written, but a bit lengthy, could be condensed. Consider e.g. shortening the first two paragraphs about primary production and climate change, for example LL58-60 could be skipped, and maybe also the following sentence about the mismatch for grazers as it is not really related to your study, and LL82-83 as you later on in the Methods section state that samples were obtained by Inuit harvesting, and/or LL71-89.

L53: '...drives the energy flow...'

L100: Consider adding phenylalanine in brackets for the essential AAs and also add the term 'source AA' as this comes up later on in the text.

L102: I'm not sure why you compare the AA-CSIA to BSIA since you didn't run bulk analysis. Maybe you only focus on the methods you actually applied, and you could skip LL102-106 (also to shorten the text).

L107: Put glutamic acid in brackets after trophic AA.

L115: I'm not sure if generalizing as 'benthic fauna' is a good idea, maybe stick to 'Atlantic walrus'.

L118: 'Predicted' sounds like you run actual models to predict something. Maybe rephrase to 'we hypothesize'. The same further down in this paragraph.

LL124-127: But wouldn't it be possible/likely to see a change in $\delta^{15}\text{N}$ as availability, concentrations of N-sources change due to environmental changes?

Methods

This section is very well and clearly written.

L 152: Can you specify in which lab the HBI samples have been analysed?

LL197-199: You could add a couple of references for this statement.

L208: Table 1 shows that there were between 2 and 5 individuals per year, suggest to also specify in the text.

L219: I don't think the term TDF has been introduced yet. I saw it has been explained in the Supporting Information, but it should also be introduced here.

L232: Student's t-test? Paired/unpaired?

L234: Consider adding the packages you used in R.

Results

I wonder if it would be better to start the Results section with the basic environmental information about sea ice concentrations and temporal changes. So, you could consider to have (a) temporal trends in sea ice concentration including LL247-257, then (b) Relationship between sea ice-derived carbon and $\delta^{13}\text{C}_{\text{Phe}}$ and (c) Temporal trends in sea ice-derived carbon, $\delta^{13}\text{C}_{\text{Phe}}$, $\delta^{15}\text{N}_{\text{Phe}}$ and trophic position.

Table 1: Is there a reason why you only present $\delta^{13}\text{CPhe}$ and $\delta^{13}\text{CGlu}$ from muscle tissue and % sea ice-derived carbon from liver tissue (the time frame of trophic information maybe?).

Consider stating in the Methods section why you used which tissue for which analysis.

L258: '...there were no significant differences..'

L258-266: You should more frequently refer to Table 1 where the actual values can be found.

Consider adding a sentence about the partly great differences in TP calculation using the different published equations. It has been explained well in the Supporting Information, but maybe you could state again that the higher estimations are likely the more realistic values.

Figure 1: Could you illustrate in the first map (A) where Foxe Basin and Jones Sound are located?

Discussion

This section is also very well written and explains the most important results of this study.

L292: You should probably say TP of 2-3 since there were large differences between the calculations, or put the range in brackets (1.9-3.2).

L318-L320: Could it also indicate a change in food sources (less ice algae-dependent prey)?

L311 and following: Can you specify from which tissue (muscle or liver) these values are derived from, so the reader can get some idea about time frames?

L322: You could probably be more specific to the season here, as ice algae during summer are likely less nutritional and attractive as a carbon source than earlier in the season.

L372-377: Here you could possibly elaborate a bit more on the differences in you TP calculations derived from the different equations. Why do you think the individuals in your study did not rely on seals? What could the stability of TP mean in terms of environmental changes, would walrus be rather resilient?

Referee: 2

Comments to the Author(s)

See file attached.

Author's Response to Decision Letter for (RSPB-2020-2126.R0)

See Appendix B.

Decision letter (RSPB-2020-2126.R1)

16-Nov-2020

Dear Dr Yurkowski

I am pleased to inform you that your manuscript entitled "Atlantic walrus signal latitudinal differences in the long-term decline of sea ice-derived carbon to benthic fauna in the Canadian Arctic" has been accepted for publication in Proceedings B.

Open Access

Paper charges

Sincerely,

Dr Daniel Costa

Associate Editor:

Board Member

Comments to Author:

Thank you for the clear and thorough addressing of all reviewer points raised.

Appendix A

Review

Atlantic walrus (*Odobenus rosmarus rosmarus*) signal a decrease in sea ice-derived carbon to benthic fauna in a melting Arctic

By David J. Yurkowski, Thomas A. Brown, Paul J. Blanchfield, Steven H. Ferguson

General

This study addresses the impacts of decline in sea ice on Arctic marine ecosystems, studying changes in the proportions of sea ice algae vs phytoplankton carbon and nitrogen sources available at the base of the marine food web. They used walrus samples, collected since 1982 to 2016 (but with many gaps in the data set) to determine if there has been a distinct change in the availability of ice algae as sea ice has declined in extent and duration at two different locations: one more cold and ice-rich than the other, but there both places have experienced a decline in sea ice and an earlier sea ice break-up. To study this, they performed compound specific stable isotope analyses of walrus muscle tissue, focusing on two amino acids glutamic acid (Glu) and phenylalanine (Phe), and on three highly-branched isoprenoids (H-Print) from walrus liver tissues. Data points are few, but still of interest since the data show distinct results despite being so sparse.

This study provides the first empirical support for a multi-decadal decline (1980s to 2010s) of sea ice-derived carbon in the benthic environment in association with sea ice phenology and cover that varies with latitude.

The work is of general interest and publishable – only a few minor comments, see below. The use of compound specific stable isotope analyses amino acid are few and a novel.

M&M

The authors mention that they have conducted compound specific stable isotope analyses on 12 different amino acids, but chose to only present results from two (Glu and Phe) without explaining why the ten others were discarded. The authors should explain briefly why in the manuscript.

Results and Figures

Short result text, but sufficient.

Figure 1 Here it would also be good to write location in the panel and not only show the map in the corner, especially for readers not so familiar with this part of the Arctic.

Discussion

Mean analytical precision was $<1\text{‰}$ for the compound specific stable isotope analyses which is low when the differences in values between 2006 and 2016 (Jones Sound) was just around 1‰ . Can the authors discuss how the low precision potentially may have impacted the conclusion in this study?

Any data on the development of the walrus populations in the areas studied? Healthy populations or in decline?

In the introduction the increase in seaweeds is mention as a consequence of decline in sea ice. Can the author briefly explain why only ice algae and phytoplankton are considered as carbon sources in the present study and not macro algae and microphytobenthos (grow shallow locations)? The bivalves walrus feed on are they primarily filter feeders or deposit feeders?

References

Ok, but doi numbers not provided in the reference list.

Appendix B

RSPB-2020-2126

Atlantic walrus signal a long-term decrease in sea ice-derived carbon to benthic fauna in the mid-latitude Canadian Arctic

We thank both reviewers and the Associate Editor for their thoughtful comments and suggestions which improved the manuscript as a whole. We have incorporated and addressed all of their comments in the manuscript and in comments below.

Associate Editor

Both reviewers assess the study positively, while including multiple suggestions for minor revision. In particular, as well as addressing the individual suggestions, the authors are advised to try and streamline the Introduction, and one reviewer in comments to Editor made the suggestion that data from individuals should be included to satisfy the 'data availability' criteria (pointing out that only mean data are included in the paper itself).

RESPONSE: As suggested by both reviewers, we have shortened the Introduction by 1 full page (from 4 to 3 pages) and it is now more concise. This also shortened our number of references by 8. We have now made the individual data available and has been deposited into the Dryad repository that will be linked with this article.

Referee: 1

This is a very nice study with some surprising results, and a very well written paper! I only have some relatively minor suggestions, and I'm looking forward to see the paper published.

RESPONSE: Thank you

Title

Would 'Atlantic walrus signal a decrease in sea ice-derived carbon to benthic fauna in the mid-latitude Canadian Arctic' be more appropriate? From your results, you found a strong decline in the utilization of sea ice-derived carbon in mid-latitude walrus, but it remained similarly high in Jones Sound. You should also consider including something like 'long-term' to highlight the strength of your dataset and this study.

RESPONSE: We agree and have now changed the title to "Atlantic walrus (*Odobenus rosmarus rosmarus*) signal latitudinal differences in the long-term decline of sea ice-derived carbon to benthic fauna in the Canadian Arctic".

Keywords: add Amino acids or replace CSIA with Amino acid-specific SIA

RESPONSE: We agree and have added "of amino acids"

Abstract

L26: Consider replacing 'integrate' with 'combine' or 'coupled'.

RESPONSE: Agreed – changed to "combined" – see line 26

LL 30-32: I'm not sure I understand the sentence. Wouldn't it suggest a 'coupling' instead of 'decoupling' when the contribution of sea-ice derived carbon decreases with decreasing sea ice?

RESPONSE: Strong coupling occurs when high amounts of sea ice-derived carbon or phytoplankton-derived carbon reaches the sea floor whereas a weak coupling or a

decoupling occurs when low amounts of sea ice-derived carbon or phytoplankton-derived carbon reaches the sea floor, which is the case here – see lines 31-32.

L36: Maybe instead of ‘benthic fauna’, you can say ‘Atlantic walrus’ since you didn’t investigate different benthic fauna in this study, but walrus that is preying on this fauna.

RESPONSE: agreed – changed to “Atlantic walrus and their benthic prey”– see lines 35-38.

How are your results related to ‘a change in Arctic marine ecosystem functioning’? Maybe you can be a bit more specific (or leave it out)?

RESPONSE: We have clarified this statement where we were inferring functioning (transferring of nutrients) between sympagic-pelagic-benthic habitats and have now included this “a change in Arctic marine ecosystem functioning between sea ice-pelagic-benthic habitats.” See lines 35-38

Specify in the Abstract that you studied samples from the Canadian Arctic, e.g. L 30 ‘... at high- and mid-latitudes in the Canadian Arctic’.

RESPONSE: Agreed – see line 30.

Introduction

The Introduction is well written, but a bit lengthy, could be condensed. Consider e.g. shortening the first two paragraphs about primary production and climate change, for example LL58-60 could be skipped, and maybe also the following sentence about the mismatch for grazers as it is not really related to your study, and LL82-83 as you later on in the Methods section state that samples were obtained by Inuit harvesting.
and/or LL71-89.

RESPONSE: Agreed – we have significantly shortened the introduction from 4 to 3 pages using many of the recommendations mentioned by this reviewer. We now have one paragraph on primary production on climate change and also shortened the paragraph on the background of Atlantic walrus (see lines 43-76).

L53: ‘...drives the energy flow...’

RESPONSE: Agreed – see line 58

L100: Consider adding phenylalanine in brackets for the essential AAs and also add the term ‘source AA’ as this comes up later on in the text.

RESPONSE: Agreed – see line 89

L102: I’m not sure why you compare the AA-CSIA to BSIA since you didn’t run bulk analysis. Maybe you only focus on the methods you actually applied, and you could skip LL102-106 (also to shorten the text).

RESPONSE: Agreed – we have removed details on bulk stable isotope analysis.

L107: Put glutamic acid in brackets after trophic AA.

RESPONSE: Agreed – see line 93.

L115: I'm not sure if generalizing as 'benthic fauna' is a good idea, maybe stick to 'Atlantic walrus'.

RESPONSE: Agreed – see line 101

L118: 'Predicted' sounds like you run actual models to predict something. Maybe rephrase to 'we hypothesize'. The same further down in this paragraph.

RESPONSE: Agreed – we changed to hypothesize. See lines 104 and 110.

LL124-127: But wouldn't it be possible/likely to see a change in $\delta^{15}\text{N}$ as availability, concentrations of N-sources change due to environmental changes?

RESPONSE: Yes, but this hypothesis is given as a null hypothesis and no change – see lines 110-113.

Methods

This section is very well and clearly written.

RESPONSE: Thank you

L 152: Can you specify in which lab the HBI samples have been analysed?

RESPONSE: Yes – we have included “Scottish Association of Marine Science”. See lines 152-153.

LL197-199: You could add a couple of references for this statement.

RESPONSE: Agreed – and we have added references [30,35] – see lines 188-190.

L208: Table 1 shows that there were between 2 and 5 individuals per year, suggest to also specify in the text.

RESPONSE: Good suggestion and we have clarified this in the text “generally 5 Atlantic walrus individuals were collected per year, except in 1987 ($N = 4$), 2013 ($N = 2$) and 2016 ($N = 2$).” See lines 198-200.

L219: I don't think the term TDF has been introduced yet. I saw it has been explained in the Supporting Information, but it should also be introduced here.

RESPONSE: Good catch – we now properly introduce this term – see lines 209-214.

L232: Student's t-test? Paired/unpaired?

RESPONSE: Yes – Student's t-test which is now mentioned - see lines 205 and 223.

L234: Consider adding the packages you used in R.

RESPONSE: The base R was used and it has been referenced properly – see line 225.

Results

I wonder if it would be better to start the Results section with the basic environmental information about sea ice concentrations and temporal changes. So, you could consider to have (a) temporal trends in sea ice concentration including LL247-257, then (b) Relationship between sea ice-derived carbon and $\delta^{13}\text{C}_{\text{Phe}}$ and (c) Temporal trends in sea ice-derived carbon, $\delta^{13}\text{C}_{\text{Phe}}$, $\delta^{15}\text{N}_{\text{Phe}}$ and trophic position.

RESPONSE: Agreed – we have moved the temporal trends in sea ice to the beginning of the results – see lines 228-239.

Table 1: Is there a reason why you only present $\delta^{13}\text{CPhe}$ and $\delta^{13}\text{CGlu}$ from muscle tissue and % sea ice-derived carbon from liver tissue (the time frame of trophic information maybe?). Consider stating in the Methods section why you used which tissue for which analysis.

RESPONSE: We have now expanded on why liver was used for HBI analysis and muscle for CSIA-AA in the Methods section – see lines 134-146.

“Based on average body mass of adult Atlantic walrus (~1000 kg), the stable isotope half-life of muscle represents approximately 6 months which encompasses part of their overwintering period in nearby polynyas, as well as spring and summer foraging periods [20]. We performed highly-branched isoprenoid diatom lipid biomarker analysis on Atlantic walrus liver samples, as >70% of highly-branched isoprenoid diatom lipids are typically stored in vertebrate liver [30]. The residency time of highly-branched isoprenoid lipids in liver of larger mammals is unknown but is thought to be days to weeks, a slightly longer timeframe to that of primary consumers (i.e. days [22,24]) yet still represents a timeframe to investigate short-term contributions. For example, Brown et al. [21] used highly-branched isoprenoids of ringed seal (*Pusa hispida*) liver samples to quantify monthly changes in sea ice-derived carbon contributions to ringed seal diet. Using both liver and muscle samples of Atlantic walrus allows one to measure time-integrated resource use over two different temporal scales (i.e. monthly using liver and seasonal using muscle).”

L258: ‘...there were no significant differences..’

RESPONSE: Changed – see line 249

L258-266: You should more frequently refer to Table 1 where the actual values can be found. Consider adding a sentence about the partly great differences in TP calculation using the different published equations. It has been explained well in the Supporting Information, but maybe you could state again that the higher estimations are likely the more realistic values.

RESPONSE: Agreed – we now refer to table 1 twice in this section. See lines 251 and 253.

Figure 1: Could you illustrate in the first map (A) where Foxe Basin and Jones Sound are located?

RESPONSE: Good suggestion – Foxe Basin and Jones Sound are now included in Fig 1.

Discussion

This section is also very well written and explains the most important results of this study.

RESPONSE: Thank you

L292: You should probably say TP of 2-3 since there were large differences between the calculations, or put the range in brackets (1.9-3.2).

RESPONSE: We prefer to mention a trophic position of 3 instead of describing variation in trophic position equations here as due to page limits – this information is provided in detail in the Supporting Information document. Equation 3 best matched that of stomach

contents and consumption of filter-feeding bivalves who are at trophic position 2 (see lines 282-285). But because of this comment, we have now briefly expanded on the differences in trophic position equations elsewhere in the Discussion (see lines 365-375 and Supplementary Information).

L318-L320: Could it also indicate a change in food sources (less ice algae-dependent prey)?

RESPONSE: Agreed – we now include “or less ice algae-dependent prey being available.” At the end of the sentence – see lines 310-314.

L311 and following: Can you specify from which tissue (muscle or liver) these values are derived from, so the reader can get some idea about time frames?

RESPONSE: We specify which tissue each analysis is from in the Methods – see lines 134-146 and have included a brief specification in the Discussion (i.e. highly-branched isoprenoid diatom lipids from liver). See line 304.

L322: You could probably be more specific to the season here, as ice algae during summer are likely less nutritional and attractive as a carbon source than earlier in the season.

RESPONSE: Agreed, we now specify the spring – see lines 314-317.

L372-377: Here you could possibly elaborate a bit more on the differences in your TP calculations derived from the different equations. Why do you think the individuals in your study did not rely on seals? What could the stability of TP mean in terms of environmental changes, would walrus be rather resilient?

RESPONSE: Agreed, see below – we have briefly expanded on differences between equations from lines 365-370. Results do not suggest ringed seal consumption due to the trophic position of Atlantic walrus (~3) being much lower than ringed seals (3.3-4.6) and this is now mentioned from lines 370-373. Referee 2 made a similar suggestion about discussing the population trend or resiliency of both Atlantic walrus populations and we have now included a statement on how both management units have a stable population size trend over time and are considered healthy populations – see lines 373-375.

Lines 365-375

“Trophic position estimates revealed that Atlantic walrus in Jones Sound and Foxe Basin have consistently played the same trophic role in each area over time. Equation (3) provided a trophic position of Atlantic walrus (range = 2.8 – 3.2) that is in better agreement with estimates derived from stomach contents (3.4) [56] compared to equations that use a universal trophic discrimination of 7.6‰ [28] which generally under-estimate the trophic position of marine mammals [36]. Though Atlantic walrus have been documented to occasionally consume ringed seals (*Pusa hispida*) [16] who occupy a trophic position of 3.3 to 4.6 [57], their contribution to Atlantic walrus diet is very low, consistent with our trophic position estimates of approximately 3. Despite declining sea ice concentration and Atlantic walrus habitat in both Jones Sound and Foxe Basin, both management units have had stable population estimates since the 1970s, suggesting healthy populations over the duration of our study period [58].”

Referee: 2

This study addresses the impacts of decline in sea ice on Arctic marine ecosystems, studying changes in the proportions of sea ice algae vs phytoplankton carbon and nitrogen sources available at the base of the marine food web. They used walrus samples, collected since 1982 to 2016 (but with many gaps in the data set) to determine if there has been a distinct change in the availability of ice algae as sea ice has declined in extent and duration at two different locations: one more cold and ice-rich than the other, but there both places have experienced a decline in sea ice and an earlier sea ice break-up. To study this, they performed compound specific stable isotope analyses of walrus muscle tissue, focusing on two amino acids glutamic acid (Glu) and phenylalanine (Phe), and on three highly-branched isoprenoids (H-Print) from walrus liver tissues. Data points are few, but still of interest since the data show distinct results despite being so sparse.

This study provides the first empirical support for a multi-decadal decline (1980s to 2010s) of sea ice-derived carbon in the benthic environment in association with sea ice phenology and cover that varies with latitude.

The work is of general interest and publishable – only a few minor comments, see below. The use of compound specific stable isotope analyses amino acid are few and a novel.

RESPONSE: Thank you

M&M

The authors mention that they have conducted compound specific stable isotope analyses on 12 different amino acids, but chose to only present results from two (Glu and Phe) without explaining why the ten others were discarded. The authors should explain briefly why in the manuscript.

RESPONSE: Agreed – we have now expanded on this point and mention that we only focus on Phe and Glu since they are the two most commonly used trophic and source amino acids (see lines 178-181).

Results and Figures

Short result text, but sufficient.

RESPONSE: We agree

Figure 1 Here it would also be good to write location in the panel and not only show the map in the corner, especially for readers not so familiar with this part of the Arctic.

RESPONSE: Agreed – we have included the locations in panels b-e.

Discussion

Mean analytical precision was $<1\text{‰}$ for the compound specific stable isotope analyses which is low when the differences in values between 2006 and 2016 (Jones Sound) was just around 1‰ . Can the authors discuss how the low precision potentially may have impacted the conclusion in this study?

RESPONSE: Good catch and we have clarified our analytical precision statements. In the previous version we referred to $<1\text{‰}$ being for both $d_{13}C$ and $d_{15}N$ for Phe and Glu but have now clarified that the precision was $\leq 0.3\text{‰}$ for $d_{13}C$ and $\leq 1\text{‰}$ for $d_{15}N$ – see line

183. The $\leq 0.3\text{‰}$ precision for $\delta^{13}\text{C}$ is much lower than the observed differences in Jones Sound (0.8 per decade) and Foxe Basin (0.6 per decade).

Any data on the development of the walrus populations in the areas studied? Healthy populations or in decline?

RESPONSE: Despite declining sea ice concentration and Atlantic walrus habitat in both Jones Sound and Foxe Basin, both management units have had stable population estimates since the 1970s, suggesting healthy populations over the duration of our study period [58].” – see lines 372-375.

In the introduction the increase in seaweeds is mention as a consequence of decline in sea ice. Can the author briefly explain why only ice algae and phytoplankton are considered as carbon sources in the present study and not macro algae and microphytobenthos (grow shallow locations)? The bivalves walrus feed on are they primarily filter feeders or deposit feeders?

RESPONSE: We have clarified that Atlantic walrus are feeding on filter-feeding bivalves (see lines 67 and 283) and therefore with these bivalve species (primarily *Mya truncata* – see Fisher & Stewart 1987 – CJZ) not being deposit feeders, macroalgae is an unlikely carbon source. The reviewer brings up a good suggestion regarding microphytobenthos, but these carbon sources are most common in very shallow water with high light penetration (<5m; Blackford 2002 – Estuarine, Coastal and Shelf Science, 55, 109-123) whereas Atlantic walrus typically dive to depths over 10m for their prey (Garde et al. 2008 – JEMBE, 500, 89-99).

References

Ok, but doi numbers not provided in the reference list.

RESPONSE: DOIs have now been provided for each reference